# A new approach to cultural scripts of trauma sequelae assessment: The sample case of Switzerland

**Rahel Bachem** *, **Amelie Mazza, David J. Eberle, Andreas Maercker**

Psychopathology and Clinical Intervention, Institute of Psychology, University of Zurich, Zurich, Switzerland

* r.bachem@psychologie.uzh.ch

**Data Availability Statement:** The data for this manuscript is qualitative and highly sensitive. To protect the confidentiality and anonymity of the participants, the interview transcripts are not shared publicly. Example participant statements

## Abstract

### Background

The novel concept of cultural scripts of trauma sequelae captures culture-specific expressions of posttraumatic distress (e.g., cognitive, emotional, interpersonal, psychosomatic changes) and their temporal associations. Cultural scripts of trauma sequelae complement pan-cultural (etic) diagnoses, such as posttraumatic stress disorder (PTSD) and Complex PTSD, as well as the cultural syndromes concept.

### Objective

This study aimed to develop the cultural scripts of trauma inventory (CSTI) for German-speaking Switzerland and to explore temporal associations of script elements.

### Method

Five semi-structured focus groups were conducted with psychotraumatologists ($n = 8$) and Swiss trauma survivors ($n = 7$). The interview schedule included open questions about different domains of potential posttraumatic changes (emotions, cognitions, worldviews, interpersonal relationships, body-related experiences, behavior, and growth). Data were analyzed using qualitative content analysis.

### Results

The Swiss CSTI includes 57 emic elements that represent salient trauma sequelae (30 conformed with a theoretically derived item pool, 27 were newly phrased). Temporal script associations were visualized in a network, whereby self-deprecation, the urge to function and overcompensate, and the urge to hide and endure suffering had the highest number of connections.

### Conclusion

While many posttraumatic changes identified in the present work seem to mirror pan-cultural phenomena represented in the Complex PTSD concept (e.g., self-deprecation), others (e.g., urge to function and perform, urge to hide and endure suffering) may be prominently

relevant to the study are shared in the supplementary materials.

**Funding:** The author(s) received no specific funding for this work.

**Competing interests:** The authors have declared that no competing interests exist.

related to Swiss culture with its value orientations. Knowledge about cultural scripts of trauma sequelae may provide a culture-specific framework that can help to understand individual experiences of distress and enable mental health practitioners to administer culturally sensitive interventions. Pending further validation, the Swiss CSTI bears the potential to advance culture-sensitive assessment of trauma sequelae.

## Introduction

Clinical psychology has come to recognize culture as an important factor that shapes people's thoughts, perceptions, feelings, and behaviors [1–3]. As such, culture is related to the phenomenology, etiology, and expression of mental disorders. When severe and potentially traumatic experiences challenge an individual's resources and worldviews, culture provides an interpretative lens to filter, structure, and understand psychological experiences. It also represents a guide to identifying what is considered painful, important, and desirable as well as normal vs. pathological [4–6]. Thus, similar traumatic events can lead to different posttraumatic symptom presentations among people from diverse cultural contexts. Symptoms can range from nightmares or emotional numbing to panic, somatic reactions, or being possessed by spirits [1, 7–9].

The International Classification of Disease (ICD-11) [10] and the Diagnostic and Statistical Manual of Mental Disorders (DSM-5) [11] attempt to represent those aspects of the posttraumatic symptomatology that many cultures share, thereby taking an etic perspective. Indeed, studies have confirmed the validity and replicability of stress response syndromes such as PTSD and CPTSD across different cultures. For example, Knefel et al. (2019) [12] investigated the replicability of CPTSD in culturally different samples using a network analysis approach (i.e., Germany, Israel, the UK, and the USA). Further studies support the cross-cultural applicability of PTSD and CPTSD in African and Chinese communities as well as among Syrian refugees [13–16]. Studies using network analysis also suggest that the interrelationship of PTSD and CPTSD symptoms could be similar across cultures [12, 16, 17].

Nevertheless, the concepts of PTSD and CPTSD focus on a limited set of symptoms that were identified as salient trauma sequelae in a Western context and have been criticized for not being representative of posttraumatic distress in non-Western countries [18, 19]. To address the systematic neglect of culture in the conceptualization of mental disorders, different cultural concepts of distress have been developed. The DSM-5 appendix, for example, includes *cultural syndromes* (i.e., clusters of symptoms, which occur and have relevance in the culture they originate from), *cultural idioms of distress* (i.e., expressions used to describe suffering), and *causal explanations* (e.g., individuals' beliefs about the origin of their symptoms) [11, 20]. Regarding trauma and stress-related disorders, the literature has described cultural syndromes such as *Ataque de nervios* ('attacks of nerves', Latin America, Latinos in the United States, Philippines), *Baksbat* ('broken courage', Cambodia), or *Ihahamuka* ('lungs without breath', East Africa) [21, 22]. It has been argued that cultural concepts of distress represent the posttraumatic reality of trauma survivors from non-western countries better than the established concepts of PTSD and CPTSD [e.g., 23].

Cultural scripts of trauma sequelae [24] are a novel concept in clinical cultural psychology that holds promise to unravel culture-specific trauma sequelae. Scripts are generally understood as cognitive schemas that organize comprehension of event-based situations [25]. For instance, when entering a restaurant, the restaurant script provides people with common knowledge about what to expect (e.g., waiting to be seated, ordering drinks, studying the

menu, etc.). This allows the social situation to be predictable, and therefore easier to navigate. Cultural scripts of trauma sequelae specifically aim to capture those dynamic processes (e.g., cognitive, emotional, interpersonal, psychosomatic processes) that can be affected by a traumatic experience. Depending on the cultural value orientations, attitudes, and expectations predominant in a culture, a symptom can have different significance and meanings, varying from attracting compassion and social support to being politically deviant or bringing shame over the individual and/or the family [1, 26, 27]. Thus, culture defines which symptoms form the vast pool of possible posttraumatic symptoms best serve explanatory and communicative purposes in a specific context. Cultural scripts are hereby believed to function as these "interpretive lenses" [26, 28].

Importantly, cultural scripts of trauma sequelae describe self-perceived or potentially perceivable mental changes over time and thus are more than a collection of individual culture-conform symptoms. The elements within a script are temporally organized and build causal chains of components that are expected to occur in a similar order [24]. However, even though temporality is important in cultural scripts of trauma sequelae, they are not necessarily unidimensional or linear processes but can be loose and sketchy, allowing for script variation [5]. Nevertheless, they have the potential to describe and explain complex emotional, cognitive, motivational, interpersonal, and behavioral changes in trauma survivors' posttraumatic reality.

Studies have confirmed that there is an association between posttraumatic symptom presentations and value orientations [29, 30]. Importantly, cultural differences in terms of shared beliefs, value orientations, and norms also exist between Western countries [31–33]. For example, if one takes Hofstede's dimensions as a basis, Switzerland scores high on the cultural dimension of masculinity, indicating a success-oriented and competitive society, whereas the Netherlands is considered a feminine society, in which it is important to keep the life/work balance and to ensure inclusiveness. Studies suggest that such cultural value orientations are linked to the etiology as well as the phenomenology of mental disorders [34, 35]. It can thus be assumed that Western countries also differ in their cultural models of normalcy and deviancy and, consequently, in the presentation of posttraumatic symptomatology.

From a clinical psychological perspective, knowledge about cultural scripts of trauma sequelae may provide a culture-specific framework that can help to understand individual experiences of distress and enable mental health practitioners to administer culturally sensitive interventions [2, 5]. Although the clinical diagnoses of PTSD and CPTSD seem to accurately describe some features of a pan-cultural trauma response, cultural concepts, such as cultural scripts of trauma sequelae, may have greater clinical utility for being "experience-near" [18, 24, 36]. The current study is embedded in a network project focused on the exploration of cultural scripts of trauma sequelae that includes five different study sites: Switzerland, Georgia, China, Israel, and East Africa [28, 37]. In the first step, we aim to identify trauma sequelae in different cultural samples and to develop assessment tools that can adequately capture the post-traumatic changes in each culture. For this purpose, trauma sequelae items are collected and consolidated into culture-specific questionnaires: the cultural scripts of trauma inventories (CSTIs). These questionnaires will prepare the ground for broader quantitative explorations of cultural scripts of trauma sequelae.

In the present study, we first sought to develop a Swiss version of the CSTI. The items of this inventory aim to represent the overall pool of emic elements that constitute the prevailing Swiss cultural scripts of trauma sequelae. Second, we aimed to develop a network-like overview of temporally ordered cultural scripts of trauma sequelae in Switzerland. The frequency of connections between network nodes will be used to represent how central a script element is in the Swiss posttraumatic context. Overall, this exploratory project will provide new insights into pathological trauma reactions and symptoms that go beyond the study of diagnostic criteria.

## Methods

### Study design and procedure

Prior to this study, a preliminary theory-based cultural scripts of trauma inventory (PRE--CSTI) had been developed by screening existing questionnaires of trauma sequelae and the literature on cultural concepts of distress. These included, among others, the Posttraumatic Cognitions Inventory [38], the Moral Injury Scale [39], the Chinese Personality Assessment Inventory [40], and the Cultural Appendix of the DSM-5 [41]. These sources were analyzed and combined into a new list that covered a broad range of self-perceptible posttraumatic changes, including common physical symptoms and interpersonal changes. This resulted in an initial pool of 63 items, grouped into six categories, including negative changes in cognitions and affects, worldviews, interpersonal consequences, embitterment, body-related phenomena, but also experiences of growth. A pilot study with Swiss university students showed good internal consistency of the subscales [28, 42]. The main purpose of this project was a cultural adaptation by identifying which items from the PRE-CSTI are prevalent among Swiss trauma survivors, which items are missing, and which items could be deleted. Henceforth, the term "cultural scripts of trauma" always refers to trauma *sequelae*.

As there is yet little empirically derived knowledge about cultural scripts of trauma in different cultural contexts a qualitative study design was selected. Qualitative methods represent a typical approach to investigating cultural phenomena from an emic perspective [43]. Emic research aims to explore culture-specific phenomena from the viewpoint of a culture's insiders, which is in line with the present study's aim to identify posttraumatic changes among the Swiss that were not theoretically derived. To this end, five semi-structured focus group interviews were conducted (i.e., two groups with trauma experts and three groups with trauma survivors). The focus group setting provided the opportunity to investigate culturally shared posttraumatic beliefs in which cultural scripts of trauma are rooted.

### Participants and procedure

All participants had to be ≥18 years old and speak German. Additionally, experts were required to be mental health professionals (e.g., psychotherapists, body therapists, psychiatric nurses) with several years of experience in treating Swiss trauma survivors. The experts were recruited in four specialized trauma treatment centers in Switzerland, and all had trauma-specific training. Trauma survivors had to have experienced any kind of traumatic life event (e.g., severe accidents, sexual abuse, war experiences, acts of violence, or other life-threatening or horrific events or series of events), be in psychotherapeutic treatment, and express trauma-related symptoms. However, a formal diagnosis of PTSD or CPTSD was not a requirement. Moreover, they had to identify as members of the Swiss culture, which was assessed based on their self-reported cultural identification as Swiss, socialization in Switzerland, and proficiency in Swiss German. Exclusion criteria were acute suicidality or psychosis. Potential patient participants were approached by psychotherapists in the professional network of the principal investigator (RB) and informed about the study. If they were interested in participation, they reached out to the first author by phone or e-mail who consequently conducted a telephone interview (approx. 30 minutes), during which the purpose and procedure of the study were discussed. No potential participants refused to participate after this interview. This resulted in a total of eight experts (5 = female, 3 = male) and seven Swiss trauma survivors in outpatient treatment (7 = female). Recruitment took place between 25.05.2022 and 26.08.2022. Participant characteristics are presented in Table 1.

Two pilot interviews were conducted with mental health professionals before data collection to ensure the comprehensibility of the questions. The focus group interviews were

**Table 1. Participant demographics.**

| Demographics of experts (*N* = 8) | | |
|---|---|---|
| | *M* (*SD*) | Range |
| Age (years) | 41.6 (0.51) | 35–66 |
| Experience in trauma treatment (years) | 11.0 (3.2) | 7–17 |
| Time spent discussing posttraumatic adjustments during sessions (%) | 46.88 (28.9) | 10–100 |
| Swiss origin of treated trauma survivors (%) | 49.38 (17.41) | 20–75 |
| | *N* | % |
| Gender | | |
| Female | 5 | 62.5 |
| Male | 3 | 37.5 |
| Place of education | | |
| Switzerland | 3 | 37.5 |
| Germany | 3 | 37.5 |
| Switzerland and Germany | 2 | 25.0 |
| Cultural background | | |
| Switzerland | 6 | 75.0 |
| Germany | 1 | 12.5 |
| Missing | 1 | 12.5 |
| Number of trauma survivors treated | | |
| approximately 20 trauma survivors | 2 | 25.0 |
| >40 trauma survivors | 6 | 75.0 |
| **Demographics of trauma survivors (*N* = 7)** | | |
| | *M* (*SD*) | Range |
| Age (years) | 41.83 (10.92) | 30–56 |
| Time since last traumatic event (years) | 26.57 (18.08) | 3–47 |
| | *N* | % |
| Gender | | |
| Female | 7 | 100 |
| Cultural background | | |
| Swiss | 7 | 100 |
| Religious affiliation | | |
| Yes | 5 | 71.43 |
| Christianity | 5 | 71.43 |
| Education | | |
| Primary/middle school, professional training | 2 | 28.57 |
| High school | 2 | 28.57 |
| Academic (university, college degree) | 3 | 42.86 |
| Status of employment | | |
| Employed | 5 | 71.43 |
| Unemployed (disability pension) | 2 | 28.57 |
| Trauma type | | |
| Interpersonal, repeated | 6 | 85.71 |
| Interpersonal, singular | 1 | 14.29 |

conducted by the first author of the manuscript, a clinical psychologist (PhD) and licensed psychotherapist. Data collection took part at the psychotherapeutic outpatient center of the University of Zurich. In preparation for the focus groups, demographic questionnaires and written informed consent forms were filled in by the participants. Expert and survivor focus

groups were conducted separately. At the beginning of the expert focus groups, the practitioners were instructed to recall three Swiss trauma survivors they had treated and their predominant posttraumatic changes to facilitate the group discussion. The semi-structured interview schedule is shared on Open Science Framework https://osf.io/akfe9/?view_only=002bc8847 a3b4cf2be90834b480144d7 and included open questions about different domains of potential posttraumatic changes to allow for flexibility in the participants' answers. In each focus group, participants were asked how trauma affected the following areas: emotions, cognitions, worldviews, interpersonal relationships, motivation, body-related experiences, and growth. The participants were also asked about their perception of how these symptoms relate to Swiss culture. Differences between the interview guidelines for experts and trauma survivors concerned only the wording of the questions and not the content. The study was approved by the ethical committee of the University of Zurich (nr. 22.2.2).

In the summer of 2022, two focus group interviews were conducted with experts and three focus group interviews were conducted with Swiss trauma survivors. The groups took place at the outpatient Centre for Psychotherapy of [blinded]. The network project requires at each study site that a minimum of four focus groups with at least 12 individuals be conducted to reach data saturation for the qualitative analysis [44, 45]. The interviews were audiotaped and lasted 90 minutes. At the end of the group discussions, the trauma survivors' well-being was assessed and information on where to receive support in case of distress emerging from the focus group interviews was provided.

## Data analysis

First, all audiotapes were transcribed whereby the quality of the transcripts was cross-checked by two project members to ensure accuracy. Second, qualitative content analysis [46] was conducted using MAXQDA [47], applying a hierarchal coding system. The three authors RB, AM, and DE coded the data. Main categories were defined according to the domains explored in the interview schedule (focused interview analysis; level 1 codes: emotions, cognitions, worldviews, interpersonal relationships, motivation, body-related experiences, behavior, growth). Based on notes taken during the interviews and first reading of the transcripts, data-driven subcodes were developed (level 2 codes). To test the developed coding system, three independent raters coded one transcript, compared the coded segments, and discussed all discrepancies. This process resulted in a finalized, refined coding system, which was subsequently applied to all remaining transcripts by two raters. Discrepancies were discussed to develop a consensus version of the coded interviews.

Further analyses were conducted by two independent raters (level 3 codes). Rater 1 applied a deductive approach whereby coded segments were compared to the PRE-CSTI items. Coded segments that did not correspond to existing CSTI items were thematically grouped and formulated as new items. Rater 2 applied an inductive approach by formulating data-driven items based on the coded segments. These data-driven items were then compared to the PRE-CSTI items to assess which items corresponded to the PRE-CSTI, whether language adaptation was needed, and which new items should be added. Discrepancies were discussed until a consensus was reached if CSTI items would be retained, rephrased, or added to the culturally tailored Swiss-CSTI. In this semi-qualitative analysis, the frequency of codes for specific experiences was considered to increase the likelihood that specific items represented cultural script elements rather than individual experiences. Further data analysis included the coding of all statements referring to temporal connections between posttraumatic change elements. Based on these coded segments, chains of posttraumatic changes were schematically summarized. Subsequently, a visual network was compiled, whereby all temporal scripts were grouped, ordered, and connected into a network of Swiss cultural scripts of trauma.

## Results

The following sections first present the results on which the item selection for the Swiss CSTI was based. Six main categories of posttraumatic changes represent the subscales of the Swiss CSTI. Second, the results of a descriptive analysis of temporally linked script elements are presented.

### Item formulation of the Swiss CSTI

All coded text segments could be assigned to the following main categories: cognitions and affects, worldviews, interpersonal consequences, motivational changes, body-related phenomena, and growth (see S1 Table, for a summary of the code structure). Overall, 30 out of 63 theoretically derived PRE-CSTI items were confirmed, and 27 new items were developed (see S2 Table, for Swiss CSTI items; see S3 Table, for deleted PRE-CSTI items).

**Cognitions and affects.**   With 23 items (8 PRE-CSTI, 15 new), cognitions and affects represent the largest category. The most prevalent subcode among Swiss trauma survivors was self-deprecation, which included diverse negative cognitions about the self (e.g., self-blame, perceiving the self as damaged, insufficient, or bad; items 7, 8). Such cognitions were generally accompanied by distressing emotions (e.g., intense feelings of shame, guilt, weakness, disgust, helplessness, or being a failure; items 3–6, 9, 11). Shame and guilt were mentioned both referring to specific traumatic experiences as well as globally, referring to the survivors' existence and needs in general. Interestingly, the Swiss trauma survivors described a strong tendency to direct anger against themselves even when initially feeling angry at others (item 12). Participants also reported high levels of anxiety related to diverse situations, such as the fear of going mad, being touched, and being at the mercy of others (item 20).

Moreover, an insecure self-concept was described as part of the trauma sequelae of Swiss trauma survivors. Hereby, the understanding of one's reactions to certain situations seemed to be impaired, particularly in the emotional domain. These insecurities could be summarized as the feeling of not knowing oneself anymore (item 1) and difficulties in understanding and/or realize own needs (item 18). It included the fear of never being able to feel normal emotions again (item 2). Being overwhelmed by emotions, not having access to emotions, or actively repressing them were other prevalent trauma consequences (items 19, 21, 23). Feelings of grief about what had happened and thoughts about no longer wanting to live were also expressed (items 10, 22).

Another relevant posttraumatic change was the urge to maintain control over one's body and emotions (item 16). It was linked with the high intensity and unpredictability of trauma survivors' emotions and was attributed to enabling them to achieve a relative sense of safety. On the behavioral level, it could be related to restricted eating or an excessive effort to perform highly in one's job. Indeed, a characteristic belief among Swiss trauma survivors was the conviction that they must work and function at all costs (item 14). Swiss trauma survivors feel a need to work particularly hard and perform particularly well (item 13) to justify their existence and to make people perceive them as sufficient. They also reported an urge to keep up a façade of normalcy to the outside world and avoid speaking about their suffering (items 15). This aligned with a tendency to trivialize their traumatic experiences, whereby they perceived their pain as nothing special (item 17).

**Worldviews.**   Swiss trauma survivors often experience the world as a dark, bad, or meaningless place (6 PRE-CSTI, no new items). More specifically, experts and patients reported a negative expectation of the future, which is reflected in the belief that nothing good can happen to them anymore (item 24). Moreover, the world is perceived as a dangerous place, which warrants being careful because one never knows what happens next (items 25, 26). The present

data suggested that this may be related to a heightened need for control and major difficulties adapting to changes (item 28). Another central topic was the general mistrust towards other people (item 29) as humanity is seen as bad, selfish, or unpredictable. Finally, it was reported that Swiss survivors often lack a sense of belonging as they question their very right to exist (item 27).

**Interpersonal consequences.** Participants reported diverse interpersonal trauma sequelae (3 PRE-CSTI, 5 new items). In line with the general mistrust in humanity, the participants stressed that relationships have been damaged or challenged by trauma (item 30). Often the mistrust extended to all people, even the most intimate relationships. Therefore, Swiss trauma survivors tend to not share their thoughts with others (item 32). As interpersonal relationships are fundamentally challenging, Swiss trauma survivors often feel most comfortable alone (item 33). They reported great difficulties in asking others for help (item 36). However, some trauma survivors also tend to enter toxic relationships, often of a romantic nature, characterized by psychological/physical violence or exaggerated dependency on the other person (item 37). They described the feeling of not being lovable and the omnipresent fear of being rejected by others when truly seen (items 34, 35). Moreover, the Swiss survivors also expressed convictions such as being unable to satisfy relationship partners' needs or being too exhausting for other people, and thus they felt like a burden to others (item 31).

**Motivational changes.** Three items (2 PRE-CSTI, 1 new) resembled the PRE-CSTI scale previously termed 'embitterment'. Participants reported a lack of meaning in life (item 39). For example, trauma survivors mentioned thinking that everything was pointless, including being alive. Moreover, they apprehended that due to their aversive life experiences, they may never be able to achieve their dreams (item 38). Finally, an intrinsic lack of energy for life and participation in society was expressed as a salient outcome of trauma among the Swiss (item 40). The scale was thus more accurately named 'motivational changes'.

**Body-related phenomena.** Somatic experiences (9 PRE-CSTI, 4 new items) describe specific pains as well as phenomena more broadly associated with the body (e.g., eating problems, self-harming behavior). Typical concrete somatic experiences include dizziness, back pain, stomach aches, headaches (items 49–52), and, more globally, pain that is probably psychological (item 55). A general sense of not feeling well in one's body was prevalent (item 54). Moreover, expert consensus described difficulties with physical intimacy among patients with and without sexual trauma (item 46). It was also mentioned that there are diverse situations in which Swiss trauma survivors no longer feel their bodies (item 53), which represented either a general sense of disconnectedness from one's body or a dissociative response when confronted with trauma triggers and difficult emotions. Consequently, basic needs such as hunger or tiredness could not be noticed (problems with eating, severe sleep disturbances; items 56, 57). Overwhelming exhaustion was also reported frequently by survivors, referring to physical tiredness and fatigue (item 45). In some cases, participants needed self-inflicted pain (e.g., in the form of self-harm or exertion of extreme sport) to become aware of their bodies. They reported self-harming behaviors or exertion through extreme sport (item 47). Finally, abuse of alcohol or other substances is sometimes chosen to cope with trauma-related distress (item 48).

**Growth.** Besides the multitude of negative posttraumatic changes, participants also reported growth (2 PRE-CSTI, 2 new items). Experiences that were primarily mentioned as an outcome of successful psychotherapy rather than the trauma itself were not included in this section. Swiss trauma survivors often develop specific personal skills such as discovering that they are stronger than they thought they were (items 41, 43). Moreover, some developed a particular interpersonal sensitivity that allows them to understand other people in need better (item 42). Finally, it was reported as characteristic to experience natural phenomena more intensely, such as the beauty of a flower or the smell of the rain, and to be able to use these experiences as a resource for emotion regulation (item 44).

## Typical Swiss scripts of trauma sequelae

Since the cultural script of trauma approach is intended to go beyond simply listing individual culture-conform symptoms, the next step was to conduct a descriptive analysis of the temporal associations of script elements. Trauma survivors and experts spontaneously reported post-traumatic changes that were interconnected in a temporal-causal manner, which could represent cultural scripts of trauma or parts of such. A total of 70 potential scripts with two to five elements were identified in the data (see S4 Table, for a compilation of potential scripts). These associations were visualized in a descriptive network of Swiss cultural scripts of trauma (Fig 1), whereby the thickness of the arrows represents how often a connection was mentioned.

The number of connections of a network node with others could be an indicator of how central a script element is in the culture-specific posttraumatic context. Table 2 reports the number of connections for every node in the qualitative network of scripts. With 30 connections, self-deprecation was the posttraumatic experience most frequently integrated into potential scripts of trauma in the Swiss context. This included Swiss CSTI items such as believing to be weak, damaged, a burden to others, or feeling like a failure (see S5 Table, for a list of network nodes and corresponding CSTI Items). The second most frequent experiences were being overwhelmed by intense emotions and the urge to function and/or (over)compensate for perceived shortcomings (20 connections each). Finally, the urge to hide and silently endure suffering also was a highly interconnected experience (12 connections). Save for items representing experiences of growth, all items constituting the Swiss CSTI could be assigned to the different nodes in the network. General patterns of script associations will be highlighted in the discussion.

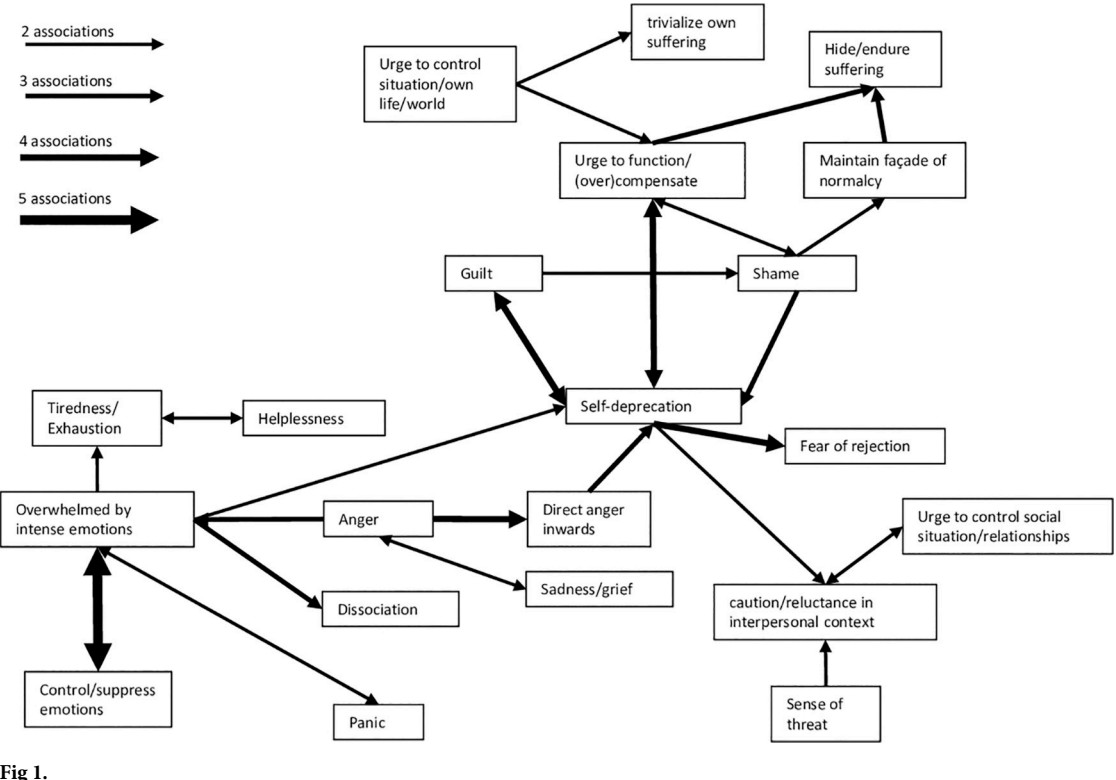

**Fig 1.**

**Table 2. MAXQDA-derived subcodes by frequency of connections for network presentation.**

| Subcodes | Number of connections |
|---|---|
| Self-deprecation | 30 |
| Overwhelmed by intense emotions | 20 |
| Urge to function/(over)compensate | 20 |
| Hide/endure suffering | 12 |
| Anger | 11 |
| Caution/reluctance in interpersonal context | 11 |
| Shame | 10 |
| Guilt | 10 |
| Control/suppress emotions | 10 |
| Urge to control social situation/relationships | 9 |
| Direct anger inwards | 8 |
| Tiredness/Exhaustion | 8 |
| Trivialize own suffering | 7 |
| Psychosomatic symptoms | 7 |
| Social withdrawal | 7 |
| Maintain façade of normalcy | 6 |
| Dissociation | 6 |
| Helplessness | 6 |
| Fear of rejection | 6 |
| Urge to control situation/own life/world | 5 |
| Unable to protect own needs | 5 |
| Mistrust | 5 |
| Sadness/grief | 4 |
| Panic | 4 |
| Not feeling one's body | 4 |
| Eating problems | 4 |
| Sense of threat | 4 |
| Negative body image | 3 |
| Substance abuse | 3 |
| Toxic relationships | 2 |
| Unable to perceive own needs | 2 |
| Self-harm/ suicidality | 2 |

## Discussion

The recent literature shows that trauma sequelae can include pan-cultural as well as culture-specific phenomena [24, 36]. The goal of the overarching network project is a better culture-specific understanding of the varying perceptions of trauma sequela among survivors from different cultural backgrounds. Such knowledge of intersubjective states of posttraumatic suffering has the potential to enrich state-of-the-art diagnostics and treatment with cultural sensitivity. In this vein, a 57-item inventory capturing salient elements of trauma sequelae among Swiss trauma survivors was developed. It consists of elements that conform with a theoretically derived item pool (PRE-CSTI) as well as 27 new elements, which are particularly likely to represent culture-specific posttraumatic changes. In an integrated discussion, we first confer whether the pan-cultural (C)PTSD symptom groups are represented among these script elements. Second, we discuss additional salient posttraumatic changes unveiled in this research and how they may relate to Swiss cultural values. Third, we explore associations in the temporal script network that stood out in terms of their centrality and frequency.

## Pan-cultural symptom groups

Many posttraumatic changes identified among Swiss trauma survivors mirrored the pan-cultural symptoms that are part of the cross-culturally validated diagnoses of PTSD and CPTSD [e.g., 15, 16]. The most prevalent and central cluster of posttraumatic cognitions and emotions can be subsumed as a negative self-concept, including posttraumatic experiences representing self-deprecation (e.g., believing to be a weak person, not feeling lovable, feeling worthless), accompanied by feelings of shame, guilt, and disgust. Moreover, the ICD-11 describes difficulties in sustaining relationships and feeling close to others as a central cross-cultural feature of CPTSD [10]. In accordance, Swiss trauma survivors were described as requiring a long time to be able to allow emotional intimacy and authentic emotional interactions with others. Notably, all trauma survivors included in the present study experienced interpersonal trauma, which is conducive to impairments in interpersonal functioning [48].

Emotion dysregulation is another core feature of ICD-11 CPTSD and has cross-cultural applicability [e.g., 15, 16]. Several posttraumatic change elements frequently mentioned for Swiss trauma survivors represent difficulties in emotion regulation (e.g., feeling overwhelmed by emotions, controlling/suppressing emotions, believing never to feel normal emotions again). Moreover, the most frequently described scripted association in the Swiss data set was the bidirectional connection between being overwhelmed by intense emotions and attempts to control/suppress emotions. This process may mirror the oscillation of re-experiencing traumatic memories, typically accompanied by strong or overwhelming emotions, and avoidance thereof, which is described for ICD-11/DSM-5 PTSD. Interestingly, the participants did not emphasize the vivid intrusive memories themselves but rather their emotional correlates.

Finally, a heightened sense of threat was reported (e.g. often feeling anxious, perceiving the world as dangerous, a need for constant alertness and caution due to the unpredictability of the future and other people), which constitutes an established symptom of PTSD. These changes in Swiss trauma survivors' worldviews are largely in line with Janoff-Bulman's (1992) [49] shattered assumption theory, which posits that a traumatic event puts into question core beliefs concerning the benevolence, meaningfulness, and predictability of the world. In conclusion, the present data included evidence of all six symptom clusters described in the symptom spectrum of ICD-11 CPTSD.

## Culture-specific symptoms & related cultural values

Above and beyond replication of these established symptoms several additional domains of posttraumatic changes were salient for Swiss trauma survivors and may be prominently related to Swiss value orientations. First and foremost, a cluster of posttraumatic cognitions and emotions was recorded, consisting of change elements related to functioning. Most of these elements represented newly phrased items based on the present data and thus may describe posttraumatic changes particularly relevant in the Swiss context. They include the belief of having to work and function at all costs, feeling the compulsion to overcompensate for own shortcomings, the urge to have control over oneself, trivializing own suffering, and having difficulties to perceive own needs. It was frequently stressed that Swiss trauma survivors tend to push themselves hard to conform with what participants experienced as central Swiss standards and expectations: being productive members of society, maintaining employment, and avoiding any dependence on social welfare institutions. These results align with Switzerland's society scoring high on the cultural dimension of masculinity, with a particular emphasis on competition, performance, and success. Similarly, Switzerland scores high on individualism, which suggests that independence, agency, and the autonomous pursuit of personal goals are highly valued [50]. Being unable to live up to the normative ideals of a competitive masculine

and individualistic society due to posttraumatic sequelae could thus be a driving force in the development of cultural scripts of trauma among Swiss survivors. In line with the present results, Austrian trauma survivors (i.e. with a similar cultural background) stated that their negative self-perceptions related to being unable to grasp chances in life and never being satisfied with their performance [48]. Supplementing these insights, a study on metaphorical concepts of resilience among Swiss trauma survivors found that their resistance and agency were embodied in the perception that overcoming adversity involves 'work' or 'taking things in hand', in order to 'function like before again' [51].

In a similar vein, several motivational changes representative of a lack of meaning, fear of never achieving one's dreams, and a lack of energy for life were discovered. These experiences are conceptually related to self-efficacy, self-worth, and motivation but are not directly included in the CPTSD symptom spectrum. Nevertheless, ample empirical evidence established an association between motivational dysfunction and PTSD (Simmen-Janevska et al., 2012, for a review) [52]. From a cultural perspective, they may result in significant psychological distress among Swiss trauma survivors as they represent a divergence from the culturally valorized models of competitive engagement and drive [5, 53].

Another potentially culture-related manifestation of posttraumatic distress pertains to emotion dysregulation, which was found to be informed by cultural norms [36]. The present findings suggest that Swiss trauma survivors tend not to express anger toward others but to direct it inward and have a tendency to self-blame (i.e., the inability to blame others). This practice seemed to be, on the one hand, related to the pronounced self-deprecation, whereby trauma survivors devalued themselves rather than the other person. On the other hand, it was perceived to be associated with a salient Swiss cultural value by which people are generally not encouraged to express anger or aggression.

Somatization and other body-related change elements are common phenomena among trauma survivors, related to both psychological and biological posttraumatic changes [54]. Experts have argued that the exclusion of somatic symptoms from the (C)PTSD diagnoses limits their validity in diverse cultural groups [18]. In accordance, many theoretically derived CSTI items were confirmed among Swiss trauma survivors (e.g. tiredness/exhaustion, psychosomatic symptoms representing diverse manifestations of pain, and dissociation of one's body or body parts). It is noteworthy, however, that body-related symptoms were mentioned relatively fewer times in the focus group discussions compared to emotional and cognitive posttraumatic change elements. The underlying reason might lie in the relative importance assigned to emotions vs. bodily sensations in Swiss culture. Research showed, for example, that depressed Chinese study participants spontaneously reported more somatic symptoms compared to Canadian participants, who focused more on psychological symptoms [55]. Consequently, it could be assumed that while experiencing somatic symptoms is a cross-cultural phenomenon, its prioritization depends on culture.

Finally, despite the predominantly pathological perspective on trauma sequelae, participants also reported experiences of posttraumatic growth. These experiences primarily represented developing interpersonal sensitivity, personal skills, and strength. The latter was also identified as a Swiss metaphor for posttraumatic growth among survivors of a devastating landslide (i.e., that the adversity had 'shaped and molded' them in such a way that they 'felt stronger') [51]. Interestingly, trauma-related spiritual or religious growth, an established dimension of posttraumatic growth [56], was not reported. The lack of responses about spirituality in a traditional sense is explicable by the progressing secularity of Switzerland [57]. However, participants frequently mentioned experiencing natural phenomena more intensely, represented as a newly phrased item in the Swiss CSTI, and one expert observed that this can be perceived by the Swiss as a spiritual experience and may represent an underestimated resource in this culture.

## Script network

The cultural script approach offers a way to capture complex and sequentially arranged models of traumatic stress and thus informs us about the dynamic posttraumatic experience of trauma survivors [24]. Even though the present analysis of script associations is descriptive, it revealed some clinically relevant trends that might be further explored in quantitative and experimental studies. For example, the qualitative script network analysis informed about central posttraumatic change elements with a particularly high number of interconnections with other changes (e.g., self-deprecation, the urge to function and overcompensate, and the urge to hide and endure suffering). Analogous to quantitative network analysis, such central posttraumatic changes could represent promising targets for clinical interventions [58].

We further found that multiple temporal script associations started with self-deprecation, associated with a personal sense of unworthiness, the conviction that other people are a source of threat, and a strong need to maintain control in social situations. Such scripts often ended with interpersonal restraint, such as fear of rejection, reluctance to engage in relationships, and withdrawal. Moreover, interpersonal trauma sequelae were in general most frequently endpoints of potential cultural scripts among Swiss trauma survivors. These observations suggests that disengagement with the social surroundings may be a strategy to interrupt cultural scripts of trauma related to a negative self-concept. From a cultural perspective, this strategy may be chosen particularly often due to the individualistic nature of Swiss culture, which celebrates independence and self-reliance [50].

In the emotional domain, emotion regulation difficulties (e.g., feeling overwhelmed by intense emotions, emotional control/suppression, dissociation) were the starting point for several scripts. Other temporally linked elements included self-deprecation, but also body-related phenomena such as tiredness/exhaustion and psychosomatic symptoms. Within the temporal scripts, body-related script elements were usually perceived as directly triggered by strong emotional activation or overwhelm, for example by anger, panic, or threat.

Interestingly, the present data provided no evidence that posttraumatic growth elements are embedded in temporal-causal script associations of negative trauma sequelae. It has been theorized that in the posttraumatic aftermath, different script tracks are possible, such as optimal script tracks, typical (normative) script tracks, and pathological script tracks [24]. Growth-related script tracks may represent another independent posttraumatic trajectory. However, the current data may not have explored growth-related trauma sequelae in sufficient detail to reveal them and future research is needed to explore them.

The present study has several limitations. First, generalizations are limited due to the specific characteristics of the sample, which included a small number of German-speaking participants. Moreover, the trauma survivors subsample included only females who had experienced interpersonal trauma. Importantly, however, the experts represented the perspectives of both genders and different trauma types. Second, the study design was cross-sectional, which restricts causal inferences beyond participants' personal opinions. Given the relevance of causality and chronology for the concept of cultural scripts of trauma sequelae, longitudinal studies or experimental designs are needed to further investigate script associations. Third, focus group interviews might be conducive to a social-desirability bias. Some posttraumatic changes might not have been discussed in a group setting. Finally, it is important to keep in mind that the findings depict the cultural scripts of treatment-seeking trauma survivors rather than Swiss trauma survivors in general.

## Conclusions

While many posttraumatic changes identified among Swiss patients mirrored the pan-cultural phenomena described in the CPTSD symptom profile, others may be prominently related to

Swiss value orientations and are not found in CPTSD criteria. It can thus be concluded that cultures in the Global North develop clinically relevant trauma sequelae that go beyond the ICD-11/DSM-5 concepts. In the Swiss case, they included change elements related to functioning and overcompensation, regulating anger by directing it inwards, motivational difficulties and, in the area of posttraumatic growth, a more conscious and intense connectedness with nature. As cultural scripts of trauma represent cultural models that may help therapists better understand individual experiences of distress, they have direct clinical implications. For example, for Swiss trauma patients, it could be particularly relevant to clarify how their perceptions about working, functioning and overcompensation relate to and conform with the norms of a masculine, individualistic society. Culturally sensitive psychoeducation could prompt patients to reevaluate their perspective on normalcy and deviancy and foster meaning-making processes [5]. In the next step, quantitative data will be collected to refine and validate the Swiss CSTI and systematically explore its association with cultural value orientations. Future publications from the overarching network project will aim to deepen the insights into cultural psychological factors in posttraumatic symptom development and expression and explore overlapping as well as culture-specific posttraumatic change elements across the globe.

## Supporting information

**S1 Checklist. Human participants research checklist.**
(DOCX)

**S1 Table. MAXQDA-derived code structure, number of coded segments, and example participant statements.**
(DOCX)

**S2 Table. Swiss cultural scripts of trauma inventory.**
(DOCX)

**S3 Table. PRE-CSTI items not included in Swiss CSTI.**
(DOCX)

**S4 Table. Compilation of the temporal-causal connections of script elements.**
(DOCX)

**S5 Table. CSTI network "nodes" and corresponding CSTI items.**
(DOCX)

## Author Contributions

**Conceptualization:** Rahel Bachem, Andreas Maercker.

**Data curation:** Rahel Bachem, Amelie Mazza.

**Formal analysis:** Rahel Bachem, Amelie Mazza, David J. Eberle, Andreas Maercker.

**Investigation:** Rahel Bachem, Amelie Mazza.

**Methodology:** Rahel Bachem, Amelie Mazza, Andreas Maercker.

**Project administration:** Rahel Bachem.

**Supervision:** Andreas Maercker.

**Visualization:** Rahel Bachem.

**Writing – original draft:** Rahel Bachem.

**Writing – review & editing:** Amelie Mazza, David J. Eberle, Andreas Maercker.

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
