## [Decision Letter · Decision Letter 0]

30 Jan 2024

PONE-D-23-41099A new approach to cultural scripts of trauma sequelae assessment: The sample case of SwitzerlandPLOS ONE

Dear Dr. Bachem,

Thank you for submitting your manuscript to PLOS ONE. After careful consideration, we feel that it has merit but does not fully meet PLOS ONE’s publication criteria as it currently stands. Therefore, we invite you to submit a revised version of the manuscript that addresses the points raised during the review process.

We look forward to receiving your revised manuscript.

Kind regards,

Stephan Doering, M.D.

Academic Editor

PLOS ONE

Journal Requirements:

Reviewers' comments:

Reviewer's Responses to Questions

**Comments to the Author**

1. Is the manuscript technically sound, and do the data support the conclusions?

Reviewer #1: Partly

Reviewer #2: Yes

2. Has the statistical analysis been performed appropriately and rigorously? 

Reviewer #1: N/A

Reviewer #2: N/A

3. Have the authors made all data underlying the findings in their manuscript fully available?

Reviewer #1: No

Reviewer #2: Yes

4. Is the manuscript presented in an intelligible fashion and written in standard English?

Reviewer #1: Yes

Reviewer #2: Yes

5. Review Comments to the Author

Reviewer #1: General remarks:

The study investigates cultural specific aspects of posttraumatic stress manifestations, which add on the diagnostic information. It extends the view on posttraumatic sequelae by considering the sociological perspective, which is an interesting approach.

Anyway, psychology, particularly clinical psychology has an empirical core and I am asking myself which purposes would serve such extension of information: the future development of the diagnostic systems? The further development of cultural sensitive psychotherapy? The development of new interventions at community level? Please be more specific and explain in the background why would be this research important. Also in the discussion/conclusion please indicate research gaps and suggest future research directions in this area.

Introduction

- With the phrasing oft he authors as the beginning „treatment… impeded by trauma survivors’ reluctance to seek therapeutic help, possibly because they do not recognize themselves in the symptoms that ICD-11 and DSM-5 describe …“. I am wondering to what extend trauma survivors actually look up the diagnostic symptoms, so that they may recognise themselves in the criteria or not. Please consider rephrasing.

- The autors state, “The suffering of many survivors stems from…“ Is this being assumed or already empirically well proved? Consider rephrasing.

Methods

- I do not understand the inclusion criteria of the participants. Only being ≥18 years and speaking German and beeing patient? Which kinds of patients? On what considerations? Certain diagnosis? No diagnostisc exclusion criteria? Traumatic antecedens? Which kind? Please explain. What about the experts? Did they already know the participants in advance? Were they blinded considering personal data on the participants?

- I really do not understand how the CSTI inventory and its items look like and which purposes such an instrument serves, which application it would have in the future and for what kind of output… In addition, it is not clear how the PRE-CSTI has been developed. It is confusing: which items all already included in the PRE-CSTI, which ones have been excluded and which one are new? I suggest a tabulation of the items.

Results and Discussion

- Both are very redundant, very long, and hard to follow. At the end I was wondering what is the key message, what do we actually learn from that? Please summarize the most important information and rewrite the text in a reader friendly way.

Reviewer #2: I have reviewed your manuscript with great interest. A number of suggestions follow, all of which I have made in the spirit of facilitating the readership's understanding and use of your paper. I hope they prove useful. Nevertheless, in my opinion, some changes and additions are necessary.

Introduction:

The first sentence in the introduction seems a bit strange to me. Is the crucial reason that people, no matter what they suffer from, recognize themselves in symptoms that are written in some guidelines or manuals? Isn't it the suffering? In my opinion, you can't ask anyone affected to recognize themselves in symptoms, and you certainly can't claim that not recognizing symptoms is the barrier to seeking help. The authors actually contradict themselves in the second sentence. I would ask them to clarify and correct this. With regard to the topic of the paper, I think the first paragraph could be omitted completely. The second paragraph leads very clearly to the content, while the first paragraph is confusing and seems like a foreign body in the otherwise very clear introduction.

Participants and procedure:

I would be interested in some more information about the mental health professionals. It is reported that a prerequisite for participation was "several years of experience". It would be interesting to know whether "only" experience was sufficient for participation, or whether the participating experts had additional trauma-specific training. Or did some have additional training, while others did not?

Table 1 shows the demographic data. In addition to the age of both groups, the gender distribution is missing.

Results

The presentation of the results in Supplementary Table 1 is very clear and concise. However, the structure of the text is somewhat confusing. A different structure would be very helpful for readers who are not familiar with the PRE-CSTI instrument. The reader will have a similar experience with the presentation of the section "Typical Swiss scripts of trauma sequelae". Despite the great complexity of the results obtained, a clearer presentation is highly desirable here.

This work makes a significant contribution to understanding the individual experience of traumatic events. This very helpful expansion to include cultural aspects of the consequences of traumatizing events makes the individual significance and its consequences for those affected even clearer. This understanding of cultural and thus also individual ways of experiencing things opens up a new but very important aspect of the treatment of traumatized people. The authors are also very reflective about the limitations of their study. All in all, this work makes a significant contribution to further improving the treatment of traumatized people on the basis of individual experience. It also provides a good basis for further research in this field.

6. PLOS authors have the option to publish the peer review history of their article (what does this mean?). If published, this will include your full peer review and any attached files.

Reviewer #1: No

Reviewer #2: **Yes: **Univ.-Doz. Dr. Thomas Beck

---

## [Author Response · Author response to Decision Letter 0]

11 Mar 2024

A new approach to cultural scripts of trauma sequelae assessment: The sample case of Switzerland

Review Comments to the Author

We thank the reviewers for their very thoughtful comments and made respective changes to the manuscript. Our answers appear below. We highlighted text changes in the main document in yellow. 

Reviewer #1: General remarks:

The study investigates cultural specific aspects of posttraumatic stress manifestations, which add on the diagnostic information. It extends the view on posttraumatic sequelae by considering the sociological perspective, which is an interesting approach.

Anyway, psychology, particularly clinical psychology has an empirical core and I am asking myself which purposes would serve such extension of information: the future development of the diagnostic systems? The further development of cultural sensitive psychotherapy? The development of new interventions at community level? Please be more specific and explain in the background why would be this research important. Also in the discussion/conclusion please indicate research gaps and suggest future research directions in this area.

Response: Thank you for these inputs and for pointing out issues to clarify. Our network project on culture-specific trauma sequelae that reach beyond the pan-cultural diagnoses such as PTSD and CPTSD does not aim to extend the diagnostic systems. Including all cultural variations in stress-response syndromes in the diagnostic systems does not seem feasible as it would lead to a very high number of diagnostic categories. 

The goal of our network project is a better understanding for the varying perceptions that trauma survivors have about their trauma sequela in different regions of the world. This has the potential to enrich state-of-the-art diagnostics and treatment with cultural sensitivity. 

We intend to pilot the development of Cultural Script of Trauma Inventories (CSTIs) in the cultural regions that are part of this network project. The CSTI follows the clinical rationale of the Cultural Formulation Interview (148) but represents a trauma-specific quantitative assessment tool. 

We hope that our work might serve as a facilitator for further cross-cultural studies in psychotraumatology. 

To highlight these points, we added to the manuscript in several places:

Introduction:

p. 6-7: “From a clinical psychological perspective, knowledge about cultural scripts of trauma sequelae may provide a culture-specific framework that can help to understand individual experiences of distress and enable mental health practitioners to administer culturally sensitive interventions (6,9). Although the clinical diagnoses of PTSD and CPTSD seem to accurately describe some features of a pan-cultural trauma response, cultural concepts, such as cultural scripts of trauma sequelae, may have greater clinical utility for being “experience-near” (3,26,37). The current study is embedded in a network project focused on the exploration of cultural scripts of trauma sequelae that includes five different study sites: Switzerland, Georgia, China, Israel, and East Africa (Blinded et al., under revision; Blinded, in print). In the first step, we aim to identify trauma sequelae in different cultural samples and to develop assessment tools that can adequately capture the post-traumatic changes in each culture. For this purpose, trauma sequelae items are collected and consolidated into culture-specific questionnaires: the cultural scripts of trauma inventories (CSTIs). These questionnaires will prepare the ground for broader quantitative explorations of cultural scripts of trauma sequelae.”

(…)

Overall, this exploratory project will provide new insights into pathological trauma reactions and symptoms that go beyond the study of diagnostic criteria.

Discussion:

p. 18: “The goal of the overarching network project is a better culture-specific understanding of the varying perceptions of trauma sequela among survivors from different cultural backgrounds. Such knowledge of intersubjective states of posttraumatic suffering has the potential to enrich state-of-the-art diagnostics and treatment with cultural sensitivity.”

p. 22: “The cultural script approach offers a way to capture complex and sequentially arranged models of traumatic stress and thus informs us about the dynamic posttraumatic experience of trauma survivors (24). Even though the present analysis of script associations is descriptive, it revealed some clinically relevant trends that might be further explored in quantitative and experimental studies.”

p. 24-25: “In the next step, quantitative data will be collected to refine and validate the Swiss CSTI and systematically explore its association with cultural value orientations. Future publications from the overarching network project will aim to deepen the insights into cultural psychological factors in posttraumatic symptom development and expression and explore overlapping as well as culture-specific posttraumatic change elements across the globe.”

Introduction

- With the phrasing oft he authors as the beginning „treatment… impeded by trauma survivors’ reluctance to seek therapeutic help, possibly because they do not recognize themselves in the symptoms that ICD-11 and DSM-5 describe …“. I am wondering to what extend trauma survivors actually look up the diagnostic symptoms, so that they may recognise themselves in the criteria or not. Please consider rephrasing.

- The autors state, “The suffering of many survivors stems from…“ Is this being assumed or already empirically well proved? Consider rephrasing.

Response: Thank you very much for pointing out the vagueness of the first paragraph. We intended to illustrate the discrepancy between Western diagnoses such as PTSD and non-Western representations/concepts of suffering from the perspective of patients. However, these arguments are further pursued later in the introduction. Following the suggestion of reviewer 2, who identified the same paragraph as problematic, we decided to omit it.

Methods

- I do not understand the inclusion criteria of the participants. Only being ≥18 years and speaking German and beeing patient? Which kinds of patients? On what considerations? Certain diagnosis? No diagnostisc exclusion criteria? Traumatic antecedens? Which kind? Please explain. What about the experts? Did they already know the participants in advance? Were they blinded considering personal data on the participants?

Response: We recruited patients who were in psychotherapy for trauma-related mental health symptoms. This inclusion criterion allowed us to identify individuals who were indeed affected by their traumatic experience(s). However, we did not request a specific trauma-related diagnosis such as PTSD or complex PTSD since we were interested in culture-specific reactions beyond (complex) PTSD. Exclusion criteria were acute suicidality or psychosis. We did not specify whether patients should be in- or outpatients but the final sample consisted of outpatients only. This information was added to the section “Participants and procedure” (p. 8-9).

As for the question of whether the experts knew the other participants in the focus groups in advance: some did know each other since there are only a few specialized trauma wards in Switzerland and they sometimes collaborate. Others had never met before. However, the participants were not informed ahead of time who the other focus group participants were. They were given the option to not state their names - but this option was not chosen. No personal data beyond the experts’ names and affiliations to specific therapy centers were mentioned in the focus groups. 

The expert and patient focus groups were conducted separately. Experts and patients did not interact within this study. This information was added on p. 10.

- I really do not understand how the CSTI inventory and its items look like and which purposes such an instrument serves, which application it would have in the future and for what kind of output… 

Response: The items of the Swiss CSTI inventory are presented in Supplementary Materials Table S2. The table also contains the information whether an item originated from the theory based PRE-CSTI or whether it was phrased newly based on the qualitative materials collected in the focus groups. We now show the reference to the table in a more prominent place (p.12). 

To better illustrate what the CSTI inventory looks like, we also added the instructions and rating scale to Supplementary Table S2.

We see the purpose and application of the CSTI in a) capturing culture-specific posttraumatic changes in Switzerland, including pan-cultural as well as culture-specific symptoms; b) providing a tool for future quantitative studies that assess cultural value orientations and posttraumatic sequelae; c) eventually enriching trauma diagnostics and trauma treatment with culturally sensitive information; d) facilitating further cross-cultural studies in psychotraumatology. 

Please also see our response to the first comment, where we elaborated on these points. 

In addition, it is not clear how the PRE-CSTI has been developed. It is confusing: which items all already included in the PRE-CSTI, which ones have been excluded and which one are new? I suggest a tabulation of the items.

Response: Thank you for pointing out the fact that the development of the PRE-CSTI was presented too briefly in the manuscript. We have now added some additional information on its development in the methods section (Study design and procedure; p. 8).

Moreover, a study protocol presenting the network project has been written and is currently under revision (reference must be blinded in the present manuscript). This study protocol will also describe the development of the PRE-CSTI as well as results from a pilot validation of the PRE-CSTI among 101 trauma-exposed Swiss participants.

Finally, we have added a table to the supplementary materials, which presents the PRE-CSTI items that were deleted for Switzerland (Supplementary Table S3).

Results and Discussion

- Both are very redundant, very long, and hard to follow. At the end I was wondering what is the key message, what do we actually learn from that? Please summarize the most important information and rewrite the text in a reader friendly way.

Response: We have taken the following steps to clarify the core messages and reduce redundancy:

Results:

- We provide introductory text which contextualizes and justifies the analyses carried out.

- We changed the subheadings of the different sections to provide improved guidance to the reader.

- In the analysis of individual script elements, we provide the numbers of the CSTI-CH items that were based on the results portrayed.

- The section describing typical Swiss scripts of trauma sequelae (i.e. the script network) was shortened significantly. To reduce redundancy, general patterns in the script network are now highlighted only in the discussion rather than in the results section as well as the discussion. 

- We simplified and shortened the text throughout the results section.

Discussion:

- We chose a fundamentally different structure to discuss the results in three sections. First, we discuss which CSTI elements correspond to pancultural symptoms described in ICD-11 PTSD and complex PTSD. Second, we discuss new elements and how they might relate to Swiss cultural values. Third, we discuss general trends in the descriptive network of temporal script associations. 

- We also shortened the text to increase reader friendliness.

- We added information about the next aims/steps within the network project.

With these major revisions we hope that we were successful in reducing redundancy as well as provide increased guidance to the reader about the key messages of the research.

Reviewer #2: 

I have reviewed your manuscript with great interest. A number of suggestions follow, all of which I have made in the spirit of facilitating the readership's understanding and use of your paper. I hope they prove useful. Nevertheless, in my opinion, some changes and additions are necessary.

Introduction:

The first sentence in the introduction seems a bit strange to me. Is the crucial reason that people, no matter what they suffer from, recognize themselves in symptoms that are written in some guidelines or manuals? Isn't it the suffering? In my opinion, you can't ask anyone affected to recognize themselves in symptoms, and you certainly can't claim that not recognizing symptoms is the barrier to seeking help. The authors actually contradict themselves in the second sentence. I would ask them to clarify and correct this. With regard to the topic of the paper, I think the first paragraph could be omitted completely. The second paragraph leads very clearly to the content, while the first paragraph is confusing and seems like a foreign body in the otherwise very clear introduction.

Response: Thank you very much for bringing this to our attention. We followed your suggestion to delete the first paragraph. 

Participants and procedure:

I would be interested in some more information about the mental health professionals. It is reported that a prerequisite for participation was "several years of experience". It would be interesting to know whether "only" experience was sufficient for participation, or whether the participating experts had additional trauma-specific training. Or did some have additional training, while others did not?

Response: Experts were recruited from the staff of four specialized trauma treatment centers in Switzerland. All of them had trauma-specific training. This information was added to the participants description on p. 8.

Table 1 shows the demographic data. In addition to the age of both groups, the gender distribution is missing.

Response: Thank you for bringing this to our attention. The gender distribution was added to the table.

Results

The presentation of the results in Supplementary Table 1 is very clear and concise. However, the structure of the text is somewhat confusing. A different structure would be very helpful for readers who are not familiar with the PRE-CSTI instrument. The reader will have a similar experience with the presentation of the section "Typical Swiss scripts of trauma sequelae". Despite the great complexity of the results obtained, a clearer presentation is highly desirable here.

Response: Thank you for highlighting the necessity to revise these sections. As the structure of the results section is closely tied to the steps of the qualitative data analysis, we could not choose a totally different structure. Nevertheless, we made a series of changes to the text, which we hope improve reader friendliness and clarity:

- Providing introductory text in the results chapters which contextualizes and explains the analyses.

- More descriptive subtitles for the different sub-sections.

- Stating the item numbers of the Swiss CSTI elements that were based on the specific results.

- Simplifying and shortening the text throughout the results section, particularly the description of the network of Typical Swiss scripts of trauma sequelae. 

This work makes a significant contribution to understanding the individual experience of traumatic events. This very helpful expansion to include cultural aspects of the consequences of traumatizing events makes the individual significance and its consequences for those affected even clearer. This understanding of cultural and thus also individual ways of experiencing things opens up a new but very important aspect of the treatment of traumatized people. The authors are also very reflective about the limitations of their study. All in all, this work makes a significant contribution to further improving the treatment of traumatized people on the basis of individual experience. It also provides a good basis for further research in this field.

Response: Thank you very much for this positive evaluation of our work.

---

## [Editor Report · Decision Letter 1]

20 Mar 2024

A new approach to cultural scripts of trauma sequelae assessment: The sample case of Switzerland

PONE-D-23-41099R1

Dear Dr. Bachem,

We’re pleased to inform you that your manuscript has been judged scientifically suitable for publication and will be formally accepted for publication once it meets all outstanding technical requirements.

An invoice for payment will follow shortly after the formal acceptance. To ensure an efficient process, please log into Editorial Manager at Editorial Manager® , click the 'Update My Information' link at the top of the page, and double check that your user information is up-to-date. If you have any billing related questions, please contact our Author Billing department directly at authorbilling@plos.org.

Kind regards,

Stephan Doering, M.D.

Academic Editor

PLOS ONE

---

## [Editor Report · Acceptance letter]

28 Mar 2024

PONE-D-23-41099R1 

PLOS ONE

Dear Dr. Bachem, 

I'm pleased to inform you that your manuscript has been deemed suitable for publication in PLOS ONE. Congratulations! Your manuscript is now being handed over to our production team.

Kind regards, 

on behalf of

Professor Stephan Doering 

Academic Editor

PLOS ONE